# Situational and Dispositional Achievement Goals and Measures of Sport Performance: A Systematic Review with a Meta-Analysis

**DOI:** 10.3390/sports12110299

**Published:** 2024-11-04

**Authors:** Marc Lochbaum, Cassandra Sisneros

**Affiliations:** 1Department of Kinesiology and Sport Management, Texas Tech University, Lubbock, TX 79409, USA; cassandra.sisneros@ttu.edu; 2Education Academy, Vytautas Magnus University, 44248 Kaunas, Lithuania

**Keywords:** competitive athletics, motivation, task mastery goal orientation and climate, ego performance goal orientation and climate

## Abstract

The purposes of this systematic review (PROSPERO ID: CRD42024510614, no funding source) were to quantify relationships between situational and dispositional dichotomous achievement goals and sport performance and explore potential relationship moderators. Published studies that reported at least one situational or dispositional achievement goal and a performance score were included. Studies without performance scores or based in a non-sport context were excluded. Information sources consisted of studies found in relevant published meta-analyses and EBSCOhost databases (finalized September 2024). The following statistics were conducted to assess the risk of bias: class-fail-safe *n*, Orwin’s fail-safe *n*, and funnel plots with trim and fill estimates. The summary statistics were *r* and *d*. Thirty studies from 1994 to 2024 met all inclusion criteria with 8708 participants from Europe, Asia, North America, and Oceania. The majority of samples were non-elite male youths and adolescents. The random-effects relationships (*r*) between task climate, 0.20 [0.14, 0.25], task orientation, 0.17 [0.12, 0.23], ego orientation, 0.09 [0.03, 0.16], and sport performance were small and significantly different (*p* < 0.05) from zero, while the ego motivational climate relationship was not, −0.00 [−0.48, 0.05]. The random-effects standard differences in means (*d*) for both the task orientation, 0.08 [0.02, 0.14], and ego orientation, 0.11 [−0.05, 0.26] were minimal in meaningfulness. Mixed-effects moderator analyses resulted in the following significant (*p* < 0.05) sub-group differences: subjective compared to objective performance measures (task orientation), elite compared to non-elite samples (task climate), and athlete-completed compared to coach-completed performance measures and performance records (task orientation). Finding only 30 studies meeting the inclusion criteria, which limited sub-group samples for moderation analyses, was the main limitation. Despite this limitation, AGT provides athletes and practitioners performance enhancement strategies. However, caution is warranted regarding relationship expectations given the small mean effect size values and the true prediction interval ranging from negative to positive, perhaps as a result of the heterogeneous samples and performance measures. A clear line of future research, considering the reviewed studies, with elite athletes is needed to verify the performance benefits of the task climate and ego orientation as well as the use of the ego goal orientation in selection decisions.

## 1. Introduction

Performance is a main outcome in achievement contexts. Thus, sport psychology researchers have published numerous individual studies with sport performance as a variable of great interest as the foundation for meta-analytic reviews [1]. Examples of long-standing sport psychology and performance research include topics such as mental practice [2], anxiety [3], and team cohesion [4]. Since Lochbaum and colleagues’ [1] publication reviewing sport psychology and performance meta-analyses, the published meta-analytic output concerned with sport psychology and performance continues to grow. For instance, the relationships between breathing techniques [5], mental toughness [6], and flow [7] with sport performance are recent published meta-analyses. The continued publication of sport psychology and performance meta-analyses illustrates the importance of understanding sport performance via sport psychology constructs and interventions.

Of the many sport psychology research domains, achievement goal theory (AGT) has been a dominant motivational framework guiding research since the 1980s [8]. The dichotomous achievement goal framework has accounted for an enormous volume of research output in competitive sport, leisure time exercise, and physical education investigations, as evidenced by published meta-analyses [9,10,11,12,13,14,15]. Even with the popularity of various achievement goal frameworks [16,17,18,19], the dichotomous achievement goal approach at the dispositional and situational levels is the framework of choice in sport contexts. Concerning the dichotomous goal orientations, Ivarsson and colleagues [20] quantified the relationship between goal orientations and soccer performance, and Harwood and his colleagues [12] have quantified motivational climate and performance (broadly defined in sport and physical education samples). Hence, in light of the omission in the meta-analysis literature of a comprehensive sport-based review of the relationship between dichotomous AGT and sport performance, we sought to search the AGT and the sport literature to quantify the dispositional and situational relationships with sport performance and explore the potential moderators of the relationships.

AGT originated from research teams working independently of one another as well as in collaboration in education [21,22,23,24,25]. AGT accounts for an individual’s predisposition for one of the goal orientations as well as situational influences are referred to as the motivational climate. Sport psychology researchers began studying and publishing on AGT in the late 1980s [26,27,28]. From a sport psychology perspective, John Nicholls’ framework guided the initial dialogue [8] and publications [28]. Briefly, AGT from Nicholls’ framework [25] assumes the individual in achievement contexts operates rationally and the individual’s decisions and resultant behaviors are guided by the dominant achievement goal. The goal of action is the demonstration of competence, and thus centralizes AGT on individual ability perceptions. Conceptions of ability, differentiated or undifferentiated, result from the source of reference. An individual’s ability conceptions define the task and ego achievement goals. The task orientation is dominant when personal improvement or mastery, which are self-referenced perceptions, are the motivating reasons for action and success or failure judgements. In contrast, when an individual defines his or her competence motivation by the demonstration of his or her superior ability and thus, other-referenced, the ego orientation is dominant. An athlete’s task and ego involvement are determined by his or her proneness to self- or other-referenced competence motivation and the athlete’s perception of the motivational climate.

Published meta-analyses on the dichotomous achievement goals and motivational climate provide extensive information supporting the AGT framework [9,10,12,29]. Concerning performance, Ivarsson and colleagues [20] quantified the relationship between dichotomous achievement goals with soccer performance from a handful of studies. The researchers reported a small relationship between task orientation (*d* = 0.28) and soccer performance. The relationship between ego goal orientation and soccer performance was minimal (*d* = 0.06). In a more comprehensive review, Lochbaum and Gottardy [19] quantified the relationship between approach–avoidance achievement goals and performance in sport, broadly defined across various performance tasks and participants. Lochbaum and Gottardy [19] reported significant relationships with both the mastery (*g* = 0.38) and performance (*g* = 0.38) approach goals and performance. Lochbaum, Sisneros, and Kazak [16] quantified the 3 × 2 approach–avoidance goal framework with several constructs, one being performance across sport, education, and the occupation literature. The approach (task *r* = 0.19, self *r* = 0.13, and other *r* = 0.15) relationships, most similar to the dichotomous task and ego goal orientations, were all significantly different from zero and small in meaningfulness.

Regarding AGT motivational climate research, Harwood and his colleagues provided data with what they termed “objective performance measures” that included measures such as win/loss percentage, cardiovascular fitness, evaluated skill level, and one-mile time across participants in competitive to non-competitive events. The relationship between the task climate and objective performance was positive (*r* = 0.25), whereas the relationship between the ego motivational climate was negative (*r* = −0.09). However, in our attempt to replicate Harwood’s findings, we did not find that all the studies in the “objective performance measures” category reported an objective performance measure. For instance, de Bruin et al. [30] and Brown and Fry [31] did not report objective performance measures, yet were listed as contributing to the overall effect size calculations.

### Study Purposes, Hypotheses, and Research Questions

In short, given the above reviewed AGT and sport findings that highlight the lack of a complete dichotomous AGT and sport performance quantitative review, the relationship between AGT constructs and sport performance requires more quantification. Thus, the purpose of this systematic review was to meta-analyze AGT and sport performance following the 2020 PRISMA statement [32] and explore the potential moderators of the overall mean relationships. We hypothesized small and significant relationships between the task orientation and both motivational climates with sport performance based on the reviewed meta-analyses. The relationship between both the ego orientation and climate and sport performance is mixed. The Lochbaum et al. [16] 3 × 2 meta-analysis and approach–avoidance results [19] suggest a positive relationship based on the other-approach and performance finding, whereas Harwood and colleagues’ [12] motivational climate results suggest a negative relationship. Concerning potential moderates, an examination of the performance measure characteristics (subjective, objective) is often examined in the meta-analysis literature [33,34,35,36], as has been the sample’s ability (i.e., elite or non-elite). For instance, Lochbaum and Sisneros [15] reported athlete level moderated the relationship between motivational climate and hedonic well-being. In short, we sought to explore the potential moderators of the AGT and sport performance relationships.

## 2. Materials and Methods

This systematic review with a meta-analysis followed The Preferred Reporting Items for Systematic Reviews and Meta-Analysis (PRISMA) [32]. Borenstein, Hedges, Higgins, and Rothstein’s Comprehensive Meta-Analyses (CMA) Version 4 program, statistical output with interpretations, and books guided the formulation, computation, and result interpretations [37,38,39]. As both authors have published meta-analyses together, we were conscientious of avoiding self-plagiarism. However, aspects such as explanations, table titles, and figure captions are difficult to reword as the meaning becomes confusing to inaccurate.

### 2.1. Study Eligibility Criteria

The inclusion criteria were as follows: (a) an AGT based dispositional or situational measure referenced to sport, (b) a performance measure in a sport context, (c) sufficient data provided for effect size calculation between at least one AGT construct and performance, and (d) manuscript published in a peer-reviewed academic journal. Beyond not containing sufficient data in a sport setting, the main exclusion criterion of reviewed studies was a non-competitive study context, such as a physical education class or leisure physical activity.

### 2.2. Information Sources and Search Strategy

As detailed in Figure 1, information sources included past studies in published meta-analyses (finalized 25 April 2024, reviewed for errors, and finalized 20 September 2024) and EBSCOhost database searches (finalized 12 February 2024, reviewed for errors, and finalized 20 September 2024). Both authors participated in reviewing existing databases and reference lists and performed separate EBSCOhost searches that included a discussion of findings and help with final decisions. Within EBSCOhost, we selected APA PsycArticles, ERIC, Psychology and Behavioral Sciences Collection, PsychINFO, and SPORTDiscus using the advanced search option allowing for terms in separate search boxes and delimiting the search period starting in 1980. The following searches were completed: motivational climate AND competitive sport* (by CS); motivational climate AND performance (by CS); motivational climate AND athletic outcome* (by CS); goal orientation OR goal orientations OR achievement goals OR achievement goal AND sport performance OR athlete performance (by ML); TEOSQ AND sport* (by ML) from 2016 to 2024; and POSQ AND sport* (by ML) from 2016 to 2024.

### 2.3. Data Retrieved and Coded

We retrieved the following data: participant characteristics (i.e., mean age, percent female, level of participation, country), study characteristics (i.e., design, time frame of AGT, and performance measurement), AGT measure characteristics (i.e., name of measure, measure reference), performance measure characteristics (name of measure, citations if available), data, and the study citation. Based on the performance measure descriptions, reference (to athlete or team), measure type (objective, subjective), and who completed the measure (athlete, coach) were included in our data extraction table. Based on the information provided about the participants, we used Lochbaum, Cooper, and Limps’ [40] athlete classification system. This classification system was based on Kyllo and Landers’ [41] and Swann and colleagues’ [42] classification systems and has been used in other publications (e.g., [15,36]).

Elite—International competitions at the highest level (e.g., Olympics), professional leagues (e.g., Premier leagues); described by authors as elite; samples >18 years of age.Advanced—College athletes in all countries, youth/adolescents in country level or professional team talent programs, and national-level competition.Intermediate—14 to 18 years of age; USA high school, club; not identified as elite or in college; in organized training and regional-level competition.Recreational—University intramural, adults on city teams not listed above at regional level or with extensive training schedules; sample mean age < 14 unless listed in a category above; below high school.Mix—Unable to determine a category or categories.

### 2.4. Study Quality Rating Scale

The Kmet et al. [43] quality system and scoring system was used. Each author rated sets of questions individually and consulted each other with questions during the rating process. Kmet and colleagues’ system allows for a score based on the questions used pertinent to each study’s design (e.g., cross-sectional, experimental) and are as follows:Question or objective sufficiently described?Design evident and appropriate to answer study question?Is the method of subject selection (and comparison group selection, if applicable) or source of information/input variables (e.g., for decision analysis) described and appropriate?Subject (and comparison group, if applicable) characteristics or input variables/information (e.g., for decision analyses) sufficiently described?If random allocation to treatment group was possible, is it described? N/A: Observational analytic studies. Uncontrolled experimental studies. Surveys.If interventional and blinding of investigators to intervention was possible, is it reported? N/A: Observational analytic studies. Surveys. Descriptive case series/reports.If interventional and blinding of subjects to intervention was possible, is it reported? N/A: Observational studies. Surveys. Descriptive case series/reports.Outcome and (if applicable) exposure measure(s) well-defined and robust to measurement/misclassification bias? Means of assessment reported?Sample size appropriate? N/A: Most surveys (except surveys comparing responses between groups or change over time)Analysis described and appropriate?Some estimate of variance (e.g., confidence intervals, standard errors) is reported for the main results/outcomes?Controlled for confounding? N/A: Cross-sectional surveys of a single group. Descriptive studies.Results reported in sufficient detail?Do the results support the conclusions?

### 2.5. Risk of Bias Statistics

The CMA program [39] provides statistics to test for the risk of bias across studies, which is often referred to as publication bias [44]. We examined the classic fail-safe *n* [44], Orwin’s fail-safe *n* [45], the funnel plot [46], and Duval and Tweedie’s trim and fill statistics [46]. The classic fail-safe *n* statistic represents the number of null samples required to change a significant value into a non-significant value. We specified the one-tailed test when we conducted the classic fail-safe *n* analysis. Orwin’s fail-safe *n* represents the potential missed studies that would move the correlation past a predetermined threshold. We chose zero as our missed study value and 0.10 or −0.10 as this is the threshold corresponding to the minimum cutoff point for a small effect size. The greater the value for the two fail-safe statistics, the greater the confidence that the result is safe from publication bias. In addition to the fail-safe statistics, funnel plots were examined. Funnel plots allow for the determination of whether the entered studies dispersed comparably on either side of the overall effect. Symmetry indicates that the retrieved studies captured the essence of all studies. For our last risk of bias across studies metric, we examined Duval and Tweedie’s trim and fill analysis. The trim and fill analyses were used to adjust for potential missing studies. Data points filled to the right increase the effect size value, whereas those filled to the left lower the effect size value. As with all the risk of biased statistics, they are dependent upon the search of studies and data entered as to their accuracy.

### 2.6. Summary Statistics, Planned Analyses, and Certainty Assessment

We anticipated the majority of our studies to report correlational data, thus, the correlation coefficient (*r*) was the reported summary statistic for these data. Based on Hill’s suggestion [47], mean difference data (e.g., selection or win/lose) were analyzed separate from the correlational data. For such data, we reported Cohen’s *d*. Regardless of the reported summary statistics, the random effects model was the logical choice. Of note, for Orwin’s *n* [45], only the fixed-effects analysis was provided in the CMA program. For interpretation, Cohen’s [48] guidelines of 0.10–0.29 as small, 0.30–0.49 as medium, and 0.50 as large were followed for *r,* and 0.20–0.49 as small, 0.50–0.79 as medium, and 0.80 as large for *d*. The following statistics were reported for our overall AGT and performance relationships: the number of samples, *r* or *d*, 95% confidence and prediction intervals, Tau-squared (*τ*^2^) and *I*-squared (*I*^2^), and publication bias statistics. Per study, only one summary statistic per AGT construct (i.e., task orientation, ego orientation, task climate, and ego climate) was reported. For instance, if a study reported multiple correlations for task orientation and sport performance, those were combined to one effect size value. To test categorical moderators, a mixed-effects model was used for the calculations. We used the remove-one study analysis provided in CMA to examine the robustness of our relationships beyond that of the two fail-safe statistics. The remove-one analysis runs the data with all studies except the first, and then all studies except the second, and so on with the resulting data and forest plot depicting the impact of each study. Last, and in line with the PRISMA guidelines, we provide a certainty rating for the main results.

## 3. Results

### 3.1. Study Characteristics

As found in Table 1, 30 studies, from 1994 to 2024, met all inclusion criteria. The 30 studies resulted in 217 effect size values, of which 119 were from the dispositional achievement goals (60 from ego orientation and 59 from task orientation) and 108 from the situational achievement climates (49 per each climate). The studies included 8708 participants (participant sample statistics: M = 290.27, SD = 470.43, range 28 to 2677) from Europe (Czech Republic, Germany, Italy, Netherlands, Norway, Portugal, Spain, UK), Asia (China, Iran, South Korea), North America (USA), and Oceania (Australia). Across the 30 studies, the participants’ mean age was 16.87 (SD = 4.33) with the majority of the reported mean sample ages <18 (*n* = 18). Of the samples reporting the sample sex makeup, 20.69% of the samples were 50% or greater female participants (M = 32.82% females). Studies reported on both individual sports athletes (e.g., tennis, swimming, gymnastics) and team sports athletes (e.g., handball, soccer, and volleyball). Based on our interpretation of participant descriptions, the athlete levels of competition included elite (*n* = 4) and non-elite (*n* = 26) samples. The most often used dispositional and motivational climate scales were the TEOSQ (*n* = 20) and the PMCSQ-2 (*n* = 11).

### 3.2. Study Quality

Concerning the quality score (see Figure 2 for details), the mean summary score (range from 0 to 1) was 0.93 (SD = 0.09). All samples were rated using items 1–4, 8, 10, 11, 13, and 14 [43]. The included experimental studies [71,75] (*n* = 2) were additionally rated with items 5, 6, 9, and 12, and of these, one applicable study was rated using item 7 [71]. Study quality ranged from 0.61 to 1.00. Therefore, the studies were of adequate quality.

### 3.3. Individual Study Data, Synthesis of Results, and Risk of Bias across Studies

Table 2 contains the summary data for our main four analyses. For each result, individual study data are located in Figure 3, Figure 4, Figure 5, Figure 6, Figure 7 and Figure 8. The trim and fill plots are located in the Appendix A). The random effect sizes were very small (*d*) to small (*r*) in magnitude and statistically different from zero for the task climate and orientation mean effect size results. The 95% confidence intervals for the task orientation and climate remained in the small effect size range, while the 95% true prediction intervals ranged from less than small (and negative) to medium in meaningfulness. Given the lack of variability in the task orientation and sport performance standard difference in mean, no prediction interval data were calculated. For the AGT ego results, the ego orientation and sport performance relationship were different from zero (*p* < 0.05) with the 95% confidence intervals ranging from less than small to small while the 95% true prediction interval ranged from small and negative to small and positive and medium in meaningfulness. Heterogeneity was high in the correlational results and non-existent to medium in the mean difference results. The bias statistics suggested that the task orientation (*r*) and ego orientation (*r* and *d*) effect sizes were slightly underestimated, while the other main results remained essentially identical.

**Table 2 sports-12-00299-t002:** Motivational climate and individual goal orientation results.

	Effect Size Statistics				Heterogeneity Statistics			Bias Statistics			
Correlate	k	ES	95% CI	95% PI	Q	*τ* ^2^	*I* ^2^	FS	Orwin	Trim/Fill	ES [95% CI]
TC (*r*)	13	0.20	0.14, 0.25	−0.00, 0.39	95.75	0.01	87.47	1971	14	0	Same
TO (*r*)	23	0.17	0.11, 0.23	−0.11, 0.43	216.51	0.02	89.83	1913	8	5R	0.21 [0.14, 0.28]
TO (*d*)	7	0.08	0.02, 0.14		4.92	0.00	0.00	11		2L	0.07 [0.01, 0.13]
EC (*r*)	13	−0.00	−0.48, 0.05	−0.11, 0.17	69.72	0.01	82.79	0	0	0	Same
EO (*r*)	22	0.09	0.03, 0.16	−0.19, 0.36	201.78	0.02	89.59	461	0	3R	0.13 [0.06, 0.19]
EO (*d*)	7	0.11	−0.05, 0.26	−0.31, 0.53	19.07	0.02	68.54	3		1R	0.13 [−0.02, 0.28]

Abbreviations: TC = task climate, TO = task orientation, EC = ego climate, EO = ego orientation, k = number of samples, ES = effect size, CI = confidence interval, PI = prediction interval, Q = Q total between statistics, *τ*^2^ = tau-squared, *I*^2^ = ratio of excess dispersion to total dispersion, FS = fail-safe number.

### 3.4. Additional Sensitivity Analyses

The remove-one study analysis gauges the impact of each included study. No one study appeared to influence each data set found in Table 2. The remove-one study analysis study level data are located in the Appendix A.

### 3.5. Moderator Analyses

Table 3 contains the results of the mixed-effects moderator analyses with at least three samples per moderator level: objective compared to subjective performance measures, athlete- or coach-completed performance measures or a recorded result, and athlete level, elite or non-elite. Mean difference results were presented for comparison, but not included in the mix-effects analysis. Significant (*p* < 0.05) differences resulted in three of the moderator analyses. Concerning task orientation and performance, the relationship was larger for subjective compared to objective measures and for athlete completed measures compared to coach or records. For task climate and performance, the relationship was greater in elite compared to non-elite athletes. Though not significant, the relationship between the ego orientation and sport performance relationship was larger for the elite (*r* = 0.23) than non-elite (*r* = 0.06) athletes. None of the other mixed-effects analyses approached traditional significance.

## 4. Discussion

At least 30 sports psychology and sports performance meta-analyses are in the published literature [1]. Absent from the literature is a comprehensive situational and dispositional achievement goal and sports performance meta-analysis. The present review quantified the task or mastery and ego or performance dispositional and situational achievement goals and sport performance across various athletes and sports. Both the dispositional results and motivational climate results are unique to the literature, given the minimal overlap with the two published meta-analyses of performance and AGT [12,20].

### 4.1. Summary of Findings

The task climate and orientation correlational relationships with sport performance are small and positive. When examining the 95% confidence intervals, practitioners should be confident that most athletes’ performances will benefit, albeit small in meaningfulness. However, casting some doubt as to the benefit of the task climate and orientation relationships with sport performance are the true prediction interval ranges. Researchers and practitioners should note that studies or applied work with these variables could result in minimal results, even with the promise of medium in meaningfulness upper 95% limits. Overall, practitioners should promote the task climate with no reservations, as the impact ranges from none to medium in meaningfulness. This finding is consistent with past task climate meta-analyses with a wide variety of constructs [12,15]. However, when considering the task orientation, we suggest caution if indeed the true prediction interval lower limit is negative, even if small in meaningfulness. It could be that unrealistic personal improvement goals moderate this relationship. Though across dispositional goal orientation meta-analyses concerning the various forms and goal orientation frameworks of the task or mastery goal orientation, the relationships with desired outcomes, emotions, behaviors, and thought patterns are facilitative with no glaring downsides [9,15,16,17,20]. Last, given the small and positive mean difference result, there appears to be little harm in using the task orientation as a selection variable, though no apparent benefit at least in the included studies.

In contrast to our task orientation/climate and sport performance certainly assessment and thoughts for practitioners and researchers, the ego construct results provide limited certainty to practitioners and researchers as the ego climate ranges from unrelated to hindering sport performance, based on the mean effect size and 95% confidence intervals. Whereas, the true prediction interval provides a different interpretation, as the upper limit is small and positive while the lower limit is small and negative. Thus, suggesting to practitioners to promote the ego climate as a benefit to performance is unwise given the 95% confidence and prediction interval ranges, though tempting with outcome-favored athletes and teams. For instance, in the recent Paris 2024 Olympics, top-ranked athletes and teams did win as expected. Though across the entirety of the Olympics, athletes and teams of slightly lower standing won, providing intriguing story lines. The ego disposition results initially provided some certainty that the relationship between the ego goal orientation and performance is positive, albeit small in meaningfulness. However, as with the relationship between the ego climate and sport, the true prediction interval was small and negative to positive and medium in meaningfulness. Concerning the mean difference results, there appears to be some potential for the ego disposition in selection given the positive and upper 95% confidence and true prediction intervals. However, both intervals crossed zero and thus, are not reliable. Thus, when examined in totality, the true relationship between athletes holding a higher ego disposition and sport performance lacks certainty. Our results are similar to those of past meta-analyses with the various AGT ego orientation and climate frameworks in that the overall effect size values are generally small to minimal in meaningfulness and many times are not statistically different from zero, hence, unreliable [9,12,15,16,17,20].

Regarding our moderator results, athlete-completed measures and subjective measures were greater in magnitude than coach-completed measures (95% confidence intervals marginally overlapped), records, and objective measures with the task orientation. The two moderator categories (performance measure type and who completed the measure) overlapped. The difference between the two categories was separating out the athletes and coaches when examining the subjective measures. The finding that athletes compared to their coaches were different is important. The implication is that success is defined by different standards, even subjectively between the athletes and their coaches. This type of difference is a potential source of athlete–coach friction. It is vital for athletes and their coaches to be “on the same page” when defining success. Hence, sport psychology practitioners could facilitate athletes and coaches being “on the same page” concerning defining success. For the ego goal orientation, the objective and subjective relationships were identical, as were athlete-rated performance and records. Overall, the results are consistent with AGT and support practitioners promoting and teaching the importance of holding a task orientation as one’s main disposition.

In addition to the performance measure type and who completed the measure moderator differences, the relationship between the task climate and performance was greater in elite compared to non-elite athletes, though both effect size values were small in magnitude. The importance of athlete level appeared in a recent motivational climate meta-analysis with measures of hedonic well-being [15], though in other sport psychology and performance meta-analyses, athlete level was not a significant moderator [35,36]. The result not reaching a traditional significance of interest was with the ego orientation and performance relationship differences between elite and non-elite athletes. Why this relationship differs is unknown. Within elite athletics, with winning or top performances being of enormous importance, elite athletes are best at managing such pressure and can use their ego disposition to enhance their performances.

### 4.2. Limitations and Future Directions

Despite following the PRISMA statement, limitations exist in our review. First, and a common barrier in systematic reviews concerns article selection, is that we searched EBSCO only using English. Even so, non-English articles do have titles, abstracts, or key terms in English and thus appear in English language searches. Thus, we included non-English articles meeting our inclusion criteria. This is a limitation given the selective nature of finding such non-English studies. For these articles, we extracted information using Google Translate, which may limit the accuracy of the translation. An additional limitation is the number of samples of 30 studies, small in comparison to a recent review with 82 articles quantifying AGT motivational climate and well-being [15]. With only 30 studies, the moderator analyses were limited due to the small number of samples and the unequal sub-groups. If our sub-groups’ categories were equal in number, then this limitation would have been minimized.

In addition to limitations from the review process, limitations from the included samples, the heterogeneity in performance measures, and timing of the AGT and performance measures relative to one another are important to note. First, all the studies relied on convenience sampling. With this limitation, though the quality of the articles was high for survey studies, all the studies from a sampling perspective were low in quality. As with the recent Lochbaum and Sisneros meta-analysis with 82 studies, there are few studies with elite samples, at least with data for quantitative analysis. Harwood and his colleagues [12] did not report athlete level in their extensive motivational climate meta-analysis. With only four elite samples, our moderator analyses were limited, though the task climate from elite compared to non-elite samples reached beyond the traditional significance level. Lochbaum and colleagues’ [9] review of 260 dispositional achievement goal studies resulted in less than 15% of samples classified as elite. Given that Lochbaum and his colleagues used the information provided by the authors rather than a classification system, it is unknown whether the 15% is an over or underestimate of elite athlete samples. As a coincidence, the four elite samples in this review are 15% of the total studies. The barrier to elite athlete research is the often-discussed lack of access. Another limitation, though partially explained by our moderator analyses, was the variety of performance measures. Performance measure heterogeneity is an issue across sport psychology meta-analyses and could contribute to wider 95% confidence interval ranges. However, the performance measure moderator results were only significant for the task goal orientation. Lastly concerning limitations found within the included studies, the measurement timing of the AGT variables in relation to performance is another limitation. Unlike previous meta-analyses with the sport psychology construct measured before the sport performance with sufficient samples for a meta-analysis [34,35,36], the AGT and sport performance literature is not such literature and thus, most of our relationships were cross-sectional.

The findings and limitations provide directions for future research directions. For researchers interested in AGT and sport performance, they should be aware that relationships may range and thus provide conflicting results across studies based partially on the performance measure. We recommend that researchers be very precise in their research questions and consistent in their samples and measures to help best understand the relationships between AGT and sport performance. Concerning a specific future research need, the most glaring need is research with Masters athletes. As mentioned by Lochbaum and Sisneros [15], motivational climate research with Masters athletes is absent and thus encouraged. In their 82 samples, not one mean age was greater than 26 years of age. Moreover, Lochbaum et al. [9] reported only 3% of the participant groups in the 260 quantified goal orientation studies included Masters athletes. Whether AGT constructs are related to Masters athletes’ performance is important in future research. Next in line for future research concerns the athlete level moderator of results. We suggest future research that examines whether the ego goal orientation does impact sport performance within samples of elite athletes. Given that elite sport environments are win orientated, the importance of the task climate needs future research. Could it be that a high task climate and ego orientation interact to result in the best performances? Last, researchers should utilize an athlete level coding system such as the one steadily developed by Lochbaum, Cooper, and Limp [40] based on Kyllo and Landers [41] and Swann and colleagues [42]. Swann and colleagues’ system is the most complete and often cited, though in our gathered studies, it is not used.

## 5. Conclusions

Understanding and improving sport performance is one aspect of sport psychology research and practice. Thus, quantifying the situational and dispositional achievement goal-based constructs and sport performance relationships are one more step towards better understanding and improving sport performance. Despite the discussed limitations, the knowledge gained from our review is valuable to academics, practitioners, coaches, and athletes. Of the most importance and in line with past AGT meta-analyzed research [9,16,29], the task orientation at the dispositional and situational level were significantly related to performance as was the task orientation as a selection variable. Though small in meaningfulness, the task climate and orientation correlational values with performance are similar in meaningfulness to other sport psychology constructs such as self-confidence, self-efficacy, and mood as well as interventions such as music and goal setting. Promotion of both is a clear real-world application of the results. Of note, the ego goal orientation effect size value was significantly different from zero. Thus, in conclusion, the quantified findings suggested the performance benefits of the task and ego goal orientations and task climate. The moderator results suggested elite athletes’ performance benefited from higher perceptions of the task climate and ego goal orientation. With this knowledge, two applied suggestions from the results are that coaches and practitioners with elite athletes should condone the “winning mindset” while practitioners at sub-elite levels may foster a task orientation and climate by encouraging personal improvement.

## Figures and Tables

**Figure 1 sports-12-00299-f001:**
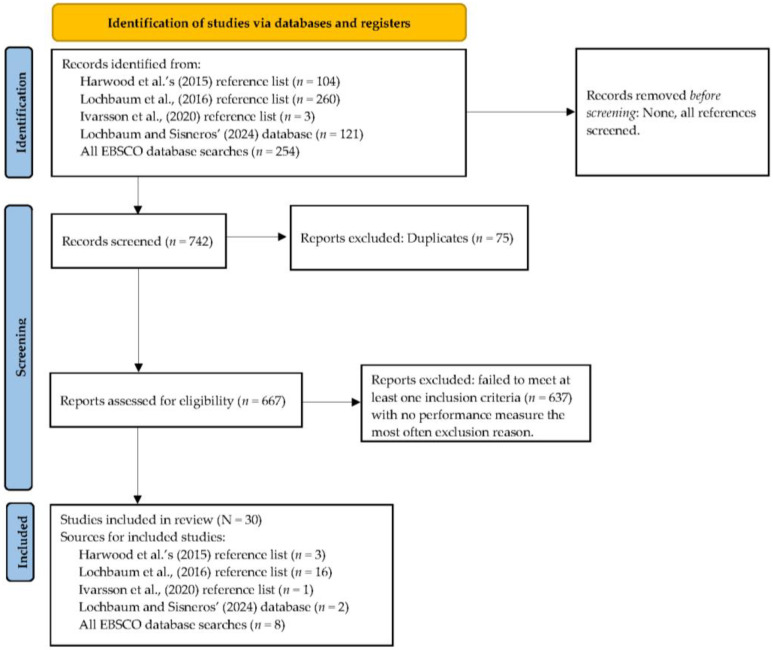
PRISMA flow chart for the identification of the included studies [9,12,15,20].

**Figure 2 sports-12-00299-f002:**
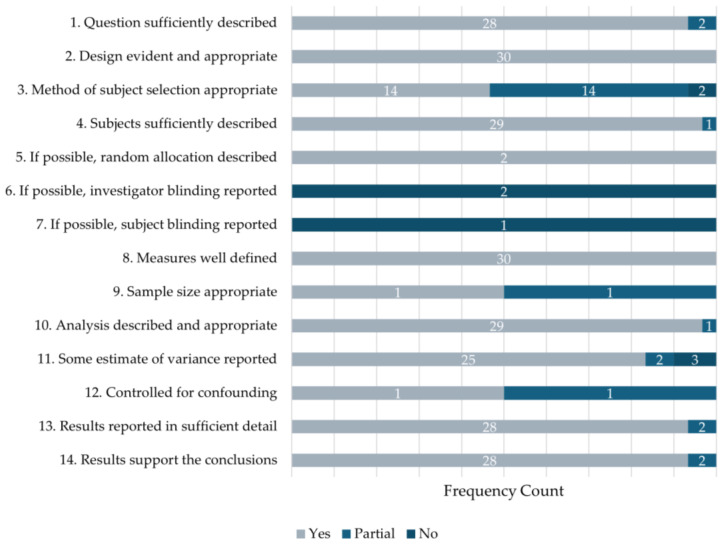
Study quality frequency counts by question.

**Figure 3 sports-12-00299-f003:**
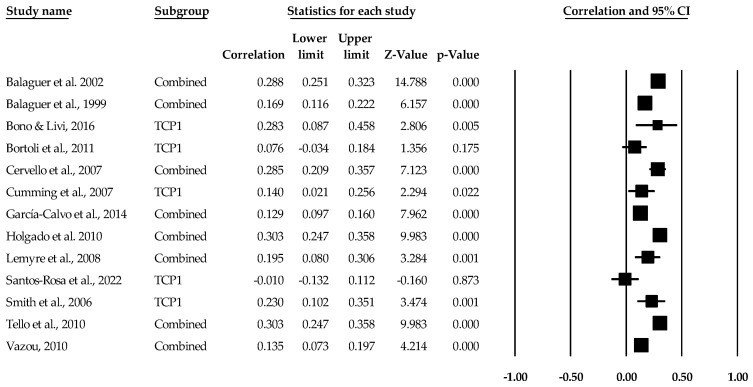
Task climate and performance statistics (correlations) with corresponding forest plots. Combined means more than one task climate and performance data point entered for the listed study. TCP1 means only one data point for the task climate and performance entered for the listed study. Figure references [50,51,52,53,56,57,63,64,70,72,73,74,78].

**Figure 4 sports-12-00299-f004:**
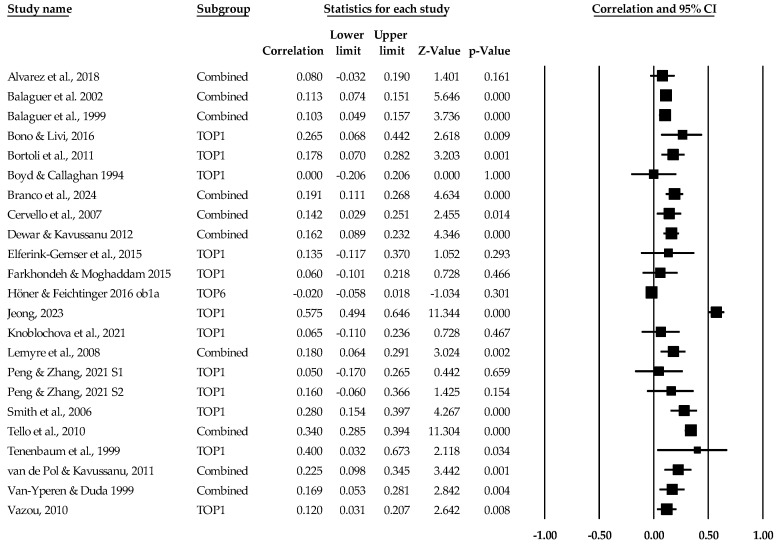
Task orientation and performance statistics (correlations) with corresponding forest plots. Combined means more than one task orientation and performance data point entered for the listed study. TOP1 means only one data point for the task orientation and performance entered for the listed study. Figure references [49,50,51,52,53,54,55,56,58,60,61,65,67,68,69,70,71,73,74,75,76,77,78].

**Figure 5 sports-12-00299-f005:**
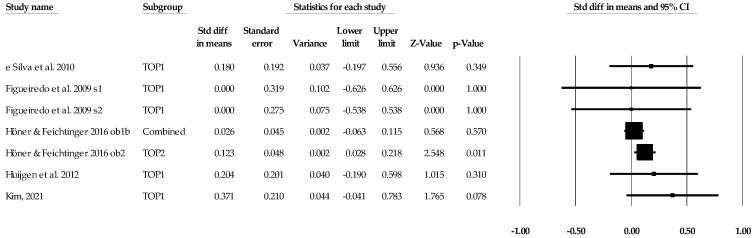
Task orientation and performance statistics (standard difference in means) with corresponding forest plots. Combined means more than one task orientation and performance data point entered for the listed study. TOP1 means only one data point for the task orientation and performance entered for the listed study. Figure references [59,62,65,66,68].

**Figure 6 sports-12-00299-f006:**
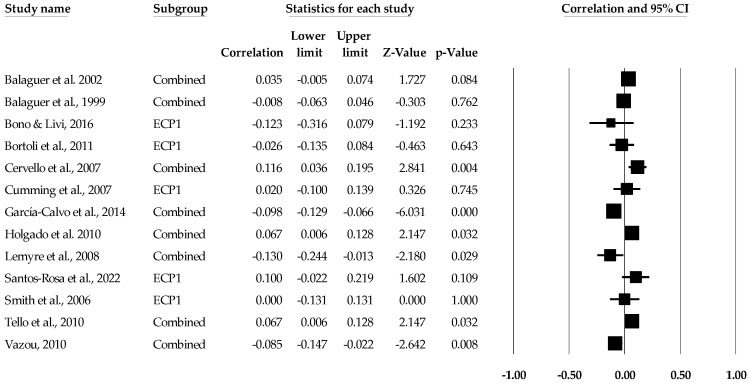
Ego climate and performance statistics (correlations) with corresponding forest plots. Combined means more than one ego climate and performance data point entered for the listed study. ECP1 means only one data point for the ego climate and performance entered for the listed study. Figure references [50,51,52,53,56,57,63,64,70,72,73,74,78].

**Figure 7 sports-12-00299-f007:**
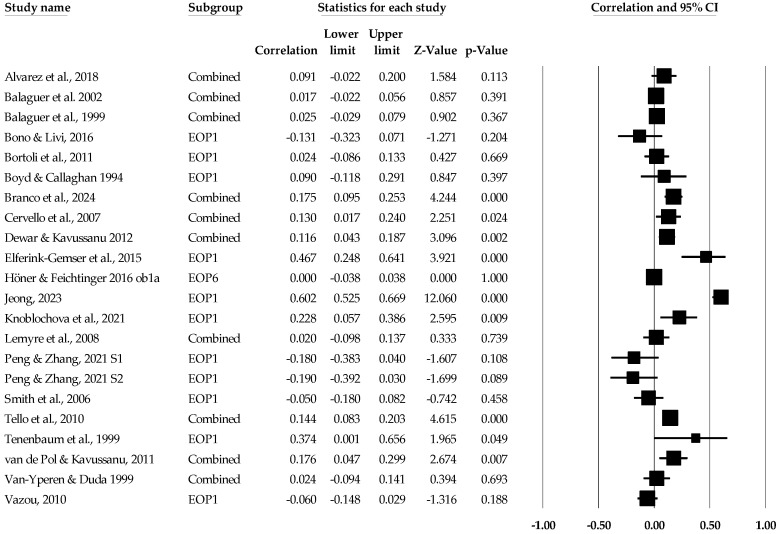
Ego orientation and performance statistics (correlations) with corresponding forest plots. Combined means more than one ego orientation and performance data point entered for the listed study. EOP1 means only one data point for the ego orientation and performance entered for the listed study. Figure references [49,50,51,52,53,54,55,56,58,60,65,67,69,70,71,73,74,75,76,77,78].

**Figure 8 sports-12-00299-f008:**
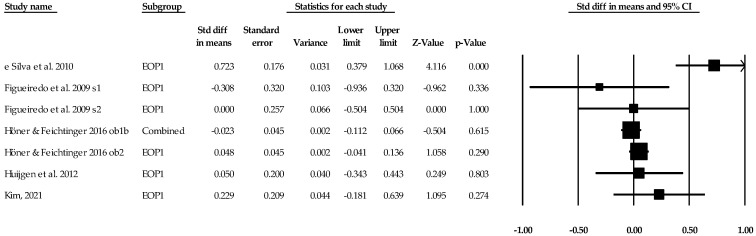
Ego orientation and performance statistics (standard difference in means) with corresponding forest plots. Combined means more than one ego orientation and performance data point entered for the listed study. EOP1 means only one data point for the ego orientation and performance entered for the listed study. Figure references [59,62,65,66,68].

**Table 1 sports-12-00299-t001:** Study characteristics.

Study	Sample and Context	Design and Data Entered	AGT	Performance
Álvarez et al. [49]	155 individual and team sport athletes (47% female, age range 18 to 36) registered on one of 13 Valencian (Spain) Community Federation teams	Retrospective, no time given, *r*	TEOSQ	Athlete self-rated performance
Balaguer et al. [50]	181 female handballers (age range 17 to 34) participating in a national competition in Spain	Retrospective thinking over the current year to date, *r*	TEOSQ, PMCSQ-2	Athlete self-rated overall performance, technical, tactical, physical, and psychological self-improvement, and performance satisfaction
Balaguer et al. [51]	219 tennis players (33% female, Mage 15.6, SD 2.1) from clubs across Spain	Retrospective thinking over the current year to date, *r*	TEOSQ, PMCSQ-2	Athlete self-rated technical, tactical, physical, and psychological self-improvement, and satisfaction with results and level of play
Bono and Livi [52]	96 swimmers (44% female, age information not provided) from 10 prestigious National Federation Rome clubs (Italy)	Retrospective thinking about past championships, *r*	TEOSQ, PMCSQ-2	Athlete self-rated performance satisfaction
Bortoli et al. [53]	320 youths (50% female, age range 13 to 14) from sport organizations and types of sports in Northeast Italy	Concurrent, *r*	TEOSQ, PMCSQ-2	Coach-rated athlete competence
Boyd and Callaghan [54]	91 male Little League baseball team members (age range, 10–12) from suburban Los Angeles (USA)	Retrospective, immediately after performance, *r*	TEOSQ	Athlete self-rated performance satisfaction
Branco et al. [55]	291 male soccer players (Mage 15.0, SD 1.5) from Portugal National Championship tournaments	Retrospective, reference to most recent performance, no time given as to most recent, *r*	AGSYS	Athlete self-rated individual and team sport performance
Cervello et al. [56]	151 tennis players (36% female, age range 12 to 16) from a major Spanish tennis development program	Prospective, one week before competition, *r*	PMCSQ-2, LAPOPECQ	Coach-rated athlete performance and athlete self-rated performance
Cumming et al. [57]	268 basketball (39% female, age range 10 to 15) players across 50 teams from 3 community centers competing in a tournament in a USA city	Retrospective, after completion of season, *r*	MCSYS	End of season won–lost percentage
Dewar and Kavussanu [58]	358 team sport athletes (34% female, age information not provided) from 26 university and 11 local teams in the UK	Retrospective, immediately after performance, *r*	POSQ	Athlete self-rated performance and athlete reported team outcome result (won, lost, draw)
e Silva et al. [59]	128 male soccer players (age range 13.0 to 14.1) from several clubs in Aveivo and Coimbra (Portugal)	Prospective or concurrent, exact time data collected before selection unclear, independent groups (mean, SD)	TEOSQ	Selection for teams for national competitions
Elferink-Gemser et al. [60]	63 speed skaters (44% female, age range 11 to 22) in a Dutch talent development program (The Netherlands)	Prospective, at start of season, *r*	TEOSQ	Performance improvement over one season based on recorded times
Farkhondeh and Moghaddam [61]	150 wrestlers (described as young, no age provided) of Ardebil Province (Iran) participating in the 2014 championship tournaments.	Prospective, before tournament, r	TEOSQ	Vague definition of successful vs. unsuccessful based on official tournament results (reported as a relationship); only task orientation relationship reported
Figueiredo et al. [62]	Initial sample of 159 soccer players (age range 11.0–14.9) from five clubs in the Portuguese midlands.	Prospective, two-year follow-up, independent groups (mean, SD)	TEOSQ	Selection of elite club teams
García-Calvo et al. [63]	377 male soccer players (age range 16 to 39) competing in the Spanish XIV group of the Third National Division	Concurrent within same time, prospective Time 0 to Time 1 and 2, Time 1 to Time 2, *r*	PMCSQ-2, PeerMCYSQ	Athlete self-rated satisfaction with participation/standing within team
Holgado et al. [64]	511 individual and team sport athletes (31% female, Mage 22.8, SD 5.2) competing in professional sports in Spain	Concurrent in nature, no time provided, *r*	POSQ, PMCSQ-2	Athlete self-rated normative success and personal improvement
Höner and Feichtinger [65]	1412 male soccer players (first data collection, all U12 classification) in a German talent development program	Concurrent in nature, odds ratio, independent groups (mean, SD), *r*	TEOSQ	Soccer motor performance assessments, coach-rated current performance, and future success defined as selection to top clubs
Huijgen et al. [66]	113 male soccer players (age range 16 to 18) from two talent development programs of professional soccer clubs (The Netherlands)	Prospective with no time provided besides “well in advance” of selection decisions, independent groups (mean, SD)	TEOSQ	Selection to continue in talent program
Jeong [67]	303 Taekwondo athletes (24% female) from 16 high schools from Seoul, Incheon, or Suwon City (South Korea)	Not stated, appears retrospective with no time provided, *r*	TEOSQ	Athlete self-rated performance
Kim [68]	92 badminton athletes (50% female, middle and high school aged) participating in the South Korean youth national team summer camp	Prospective, no exact time provided in relation to when competitions began, independent groups (mean, SD)	TEOSQ	Grouping into upper and lower performers based on camp competitions
Knoblochova et al. [69]	128 beach volleyball players (26% female, age range 14 to 42) competing in Czech national level tournaments	Prospective, *r*	POSQ	Average ranking points per tournament per athlete
Lemyre et al. [70]	141 Norwegian individual sport athletes (43% female, age range 17 to 32) from the Winter Olympic team or national sport academy	Prospective, collected during training camps/no exact time before competitions given, *r*	POSQ, PMCSQ	Athlete self-rated performance satisfaction and goal attainment over the season
Peng and Zhang [71]	78 basketball players (49% female, Mage 20.2, SD 2.6) on Chinese college basketball teams	Prospective, no exact time provided, *r*	TEOSQ	Athlete free-throw accuracy (% made)
Santos-Rosa et al. [72]	258 female gymnasts (age range 14 to 20) from 29 clubs competing in the team Spanish National Championships	Retrospective, 5 min after performance, *r*	PMCSQ-2	Athlete self-rated and coach-rated performance combined into one measure
Smith et al. [73]	223 male soccer players (age range 9 to 12) from a Valencian youth league (Spain)	Retrospective, completed month 5 or 6 of 10-month season, *r*	TEOSQ, PMCSQ-2	Athlete self-rated performance satisfaction
Tello et al. [74]	511 individual and team sport athletes (31% female, age range 16 to 45) from federations across Spain	Retrospective, no exact time provided, *r*	TEOSQ, PMCSQ-2	Athlete self-rated normative success and personal improvement
Tenenbaum et al. [75]	28 female runners (age range 13 to 16) in Australian secondary-schools training for a cross-country competition	Prospective, 4 weeks before start of training, *r*	TEOSQ	Athlete time improvement
van de Pol and Kavussanu [76]	116 tennis players (19% female, age range 16 to 40) from 28 clubs from 16 counties of Great Britain	Retrospective, over the last year and matches contributing to current rating/ranking, *r*	POSQ	Athlete self-rated performance in competition and improvement in practice combined into one measure and Lawn Tennis Association rating
Van-Yperen and Duda [77]	75 male soccer players (majority in high school, few in college) enrolled in one of five AFC Ajax schools in the Netherlands	Retrospective, no exact time provided, *r*	TEOSQ	Coach-rated and athlete self-rated performance
Vazou [78]	483 individual and team sport athletes (26% female, age range 12 to 17) from school, club, and county teams in the UK	Retrospective, last 6 months, *r*	PeerMCYSQ, PMCSQ-2, TEOSQ	Coach-rated team success in league play

AGT questionnaire abbreviations: TEOSQ = task and ego orientation in sport questionnaire; PMCSQ-2 = perceived motional climate in sport questionnaire second version; AGSYS = achievement goal scale youth sport; LAPOPECQ = learning and performance oriented in physical education climate questionnaire; MCSYS = motivational climate scale for youth sport; POSQ = perceptions of success questionnaire.

**Table 3 sports-12-00299-t003:** Moderator results.

AGT Construct	Group	k	ES	95% CI	95% PI	Q	*p*-Value
Task Orientation	Objective (*r*)	9	0.08	0.01, 0.16	−0.12, 0.28		
	Subjective (*r*)	16	0.21	0.10, 0.20	−0.13, 0.41	5.87	0.015
	Subjective (*d*) ^#^	5	0.04	−0.04, 0.12			
Ego Orientation	Objective	8	0.10	−0.02, 0.22	−0.28, 0.45		
	Subjective	16	0.10	0.02, 0.17	−0.22, 0.39	0.01	0.907
	Subjective (*d*) ^#^	5	0.11	−0.20, 0.43	−0.98, 1.22		
Task Orientation	Elite	4	0.25	−0.00, 0.47	−0.75, 0.90		
	Sub-elite	19	0.16	0.09, 0.22	−0.11, 0.40	0.53	0.465
	Sub-elite (*d*) ^#^	7	0.08	0.02, 0.14			
Task Climate	Elite	3	0.28	0.24, 0.32	−0.11, 0.60		
	Sub-elite	10	0.17	0.12, 0.23	−0.02, 0.35	9.52	0.002
Ego Orientation	Elite	4	0.23	−0.08, 0.51	−0.87, 0.95		
	Sub-elite	20	0.06	0.02, 0.11	−0.11, 0.23	1.17	0.280
	Sub-elite (*d*) ^#^	7	0.11	−0.05, 0.26	−0.31, 0.53		
Ego Climate	Elite	3	0.01	−0.07, 0.09	−0.72, 0.73		
	Sub-elite	10	−0.00	−0.06, 0.05	−0.19, 0.19	0.04	0.834
Task Orientation	Athlete	14	0.23	0.15, 0.30	−0.08, 0.50		
	Coach	4	0.12	0.07, 0.18			
	Record	9	0.08	0.01, 0.16	−0.12, 0.28	7.30	0.026
	Coach (*d*) ^#^	5	0.04	−0.04, 0.12			
Ego Orientation	Athlete	14	0.11	0.03, 0.19	−0.23, 0.42		
	Coach	4	0.02	−0.07, 0.11	−0.30, 0.34		
	Record	8	0.10	−0.02, 0.22	−0.28, 0.46	2.35	0.308
	Coach (*d*) ^#^	6	0.10	−0.07, 0.26	−0.38, 0.57		

Abbreviations: k = number of samples, *r* = mean-random-effect-modeled effect size, CI = confidence interval, PI = prediction interval, Q = Q total between statistics. ^#^ data presented, not in the mixed-effects analysis. Prediction interval data not presented when k < 6.

## Data Availability

All analyzed data are found within this article and the corresponding Appendix A.

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
