# Peer review of "Situational and Dispositional Achievement Goals and Measures of Sport Performance: A Systematic Review with a Meta-Analysis"

_sports, 2024, doi:10.3390/sports12110299_

Round 1
Reviewer 1 Report
Comments and Suggestions for Authors
This manuscript presents a review of the relationships between achievement goals and performance, and potential moderators, in sport. It is an interesting study that can contribute to the literature. However, based on what I have reviewed so far, there is a major concern about the search. At present, what is presented is not replicable and the total number of papers yielded is far too small. Based on this, I have suggested that the authors review the search and how it's being presented (see specific comments below). Until there is more information on the search (and a more thorough search undertaken), it's hard to adjudicate on what follows.
Also, there needs to be more information on the issues with the previous Harwood review and why the current review offers something new/different.
Specific comments.
L13: should 27 be twenty seven when it’s starting a sentence?
Abstract: please review the guidance for the PRISMA item 2 - https://static1.squarespace.com/static/65b880e13b6ca75573dfe217/t/65d816ce41333a04c7e1cade/1708660430868/PRISMA_2020_abstract_checklist.pdf. Reference should be made to some further information in the abstract to conform with this.
L15 remove “the”. I would say similar for 19, 22, and 25.
For the r-values, here and throughout, the 0 before the decimal can be removed – it’s not needed when the number cannot exceed 1 (unless this journal states differently to APA).
17: comma after zero.
21 and 22: hyphen needed before completed in both instances. Put the r value after completed in each case. See further examples lower down.
L15-28: this section is quite dense and I wonder if there is a way of communicating this information more succinctly. Perhaps moving the r values will help though as it’s a little difficult to follow at present.
I suggest another read through the writing in the abstract as the wording/phrasing could be smoother in a few places.
L35: to claim that “Performance is the gold standard outcome in achievement contexts” seems to be quite a stretch. Even in contexts where performance is valued, the importance of wellbeing, personal growth etc. are widely recognised. Opening with this line overlooks this, and the focus placed by sport psychologists on nurturing people as well as performers.
L36: avoid anthropomorphisms. It’s a priority of sport psychology researchers, yes.
L37-38: shouldn’t it be systematic review and meta-analysis of….
L37-38: add the year from Lochbaum’s review, albeit I don’t think the sentence needs this first line at present.
L43: suggest illustrating rather than verifying.
48-49: as evidenced x 2 is repetitive.
L50, comma needed after [19].
L53: rather than stating “in light of the surprising omission in the”, it would be more valuable to focus on what filling this gap would offer rather than stating that its omission is “surprising”.
L58-61, references needed.
L61-62: grammar issues here. Suggest rewriting.
L63: another anthropomorphism here – AGT was certainly taken up and used by researchers and should be worded as such.
L64: a rather than the.
L85-86: “using 85 quantitative methods tested” should be “used quantitative methods to test…”
L88: too much punctuation and this sentence needs restructuring.
89: should this be “the relationship between dichomotous….” The addition of relationship is also needed in 92.
89: how were the samples limited? Add more information.
89-91; this sentence could also benefit from restructuring.
L105: review the “from 0 and small” here. it’s not clear what this means.
107-109: a reference is needed to support this. I
109: what was the rational that Harwood and colleagues gave? I think this needs more unpacking.
113: the word “between” needs to be included. I also suggest this be reviewed in prior paragraphs.
116: meta-analyses should be meta-analyse.
119: add s after climate.
123: add reference.
126: “as has been the sample elite level” needs to be reviewed and made grammatically correct.
157-176: some of the information here is in the figure. Also, the remainder could simply be written as a few sentences. There should also be an appendix with the search terms applied to each data bases. The search also seems rather peculiar and only retrieving 24 papers in a search is too small. I suggest that the full search be re-run. For example, if search 2-5 are run in SportsDiscus, with OR as the Boolean operator, there are over 15000 records.
Given my concerns about the search, I suggest this needs to be undertaken again, with a table outlining the specific search (including fields searched) for each database. There is also no information on how the screening took place and who did this.
208-232: the questions don’t need to be covered, though I would expect to see an supplementary file outlining the agreed rating for each item for each study.
On the basis of the concerns surrounding the search, I suggest that this be reviewed and revised prior to any further assessment of the manuscript take place.
Comments on the Quality of English LanguageBefore re-submitting, please undertake a thorough proof read. There were lots of errors in the introduction and methods sections.
Reviewer 2 Report
Comments and Suggestions for Authors
First, I would like to recognize the authors for the systematic review and meta-analysis they conducted. The inclusion of the PRISMA checklist, CMA data file and the supplementary figures was excellent. Furthermore, the inclusion of a large number of studies (27) and a large number of participants (from European and non-European countries) were significant.
The abstract is clearly presented.
The introduction and summary are straightforward and can be easily understood even by sports scientists who are not sports psychology experts.
The purpose, hypothesis and research questions are clearly stated.
The systematic review and meta-analysis were presented in great detail.
The results section is excellent
Findings and limitations are clearly presented leading to future recommendations.
Reviewer 3 Report
Comments and Suggestions for Authors
The review is comprehensive, well-written, and organized. The purposes of this study were clear, and novel. The procedures were well described and the methods were appropriate. The manuscript was theoretically guided but lacks application of study results to real-world situations to some extent. Attempting to improve its application would be helpful.
